# Inter-sleep stage variations in corrected QT interval differ between obstructive sleep apnea patients with and without stroke history

**Serajeddin Ebrahimian**[1,2]*, **Saara Sillanmäki**[3], **Salla Hietakoste**[1,2], **Brett Duce**[4,5], **Antti Kulkas**[1,6], **Juha Töyräs**[1,7,8], **Timo Leppänen**[1,2,7], **Jukka A. Lipponen**[1,9], **Samu Kainulainen**[1,2]

**1** Department of Applied Physics, University of Eastern Finland, Kuopio, Finland, **2** Diagnostic Imaging Center, Kuopio University Hospital, Kuopio, Finland, **3** Department of Clinical Physiology and Nuclear Medicine, Kuopio University Hospital, Kuopio, Finland, **4** Sleep Disorders Centre, Department of Respiratory and Sleep Medicine, Princess Alexandra Hospital, Brisbane, Australia, **5** Institute for Health and Biomedical Innovation, Queensland University of Technology, Brisbane, Australia, **6** Department of Clinical Neurophysiology, Seinäjoki Central Hospital, Seinäjoki, Finland, **7** School of Information Technology and Electrical Engineering, The University of Queensland, Brisbane, Australia, **8** Science Service Center, Kuopio University Hospital, Kuopio, Finland, **9** Department of Emergency Care, Kuopio University Hospital, Kuopio, Finland

* serajeddin.ebrahimian@uef.fi

**Data Availability Statement:** Data cannot be shared publicly because of potentially identifying or sensitive patient information. These ethical

## Abstract

Obstructive sleep apnea (OSA) is related to the progression of cardiovascular diseases (CVD); it is an independent risk factor for stroke and is also prevalent post-stroke. Furthermore, heart rate corrected QT (QTc) is an important predictor of the risk of arrhythmia and CVD. Thus, we aimed to investigate QTc interval variations in different sleep stages in OSA patients and whether nocturnal QTc intervals differ between OSA patients with and without stroke history. 18 OSA patients (apnea-hypopnea index (AHI)$\geq$15) with previously diagnosed stroke and 18 OSA patients (AHI$\geq$15) without stroke history were studied. Subjects underwent full polysomnography including an electrocardiogram measured by modified lead II configuration. RR, QT, and QTc intervals were calculated in all sleep stages. Regression analysis was utilized to investigate possible confounding effects of sleep stages and stroke history on QTc intervals. Compared to patients without previous stroke history, QTc intervals were significantly higher ($\beta = 34$, $p<0.01$) in patients with stroke history independent of age, sex, body mass index, and OSA severity. N3 sleep ($\beta = 5.8$, $p<0.01$) and REM sleep ($\beta = 2.8$, $p<0.01$) increased QTc intervals in both patient groups. In addition, QTc intervals increased progressively ($p<0.05$) towards deeper sleep in both groups; however, the magnitude of changes compared to the wake stage was significantly higher ($p<0.05$) in patients with stroke history. The findings of this study indicate that especially in deeper sleep, OSA patients with a previous stroke have an elevated risk for QTc prolongation further increasing the risk for ventricular arrhythmogenicity and sudden cardiac death.

restrictions are imposed by the Institutional Human Research Ethics Committee of the Princess Alexandra hospital. Data are available from the Institutional Human Research Ethics Committee of the Princess Alexandra Hospital (contact via MSH-Ethics@health.qld.gov.au) for researchers who meet the criteria for access to confidential data. Researchers can contact the IHREC of PA Hospital and project steering committee will review the requests.

**Funding:** This work was supported by the European Union's Horizon 2020 Research and Innovation Programme (965417 to T.L.); The Research Committee of the Kuopio University Hospital Catchment Area for the State Research Funding (5041804 to S.K., 5041798 to S.S., 5041790 S.H., 5041794 T.L., 5101137 to J.L. and 507T044 to J.L.); The Research Foundation of the Pulmonary Diseases (to S.K. and S.H.); The Academy of Finland (323536 to T.L.); Seinäjoki Central Hospital, the Competitive State Research Financing of Expert Responsibility Area of Tampere University Hospital (VTR3242 to A.K., VTR3249 to A.K., VTR 3256 to A.K., and EVO2089 to A.K.); NordForsk (90458 to T.L. and J.T.) via Business Finland (5133/31/2018 to T.L. and J.T.); the Finnish Cultural Foundation—Pohjois-Savo regional fund and Central Fund (to S.K.); Tampere Tuberculosis Foundation (to S.K. and A.K.); the Maud Kuistila Memorial Foundation (to S.K.); Instrumentarium Science Foundation (to S.H.); Päivikki and Sakari Sohlberg Foundation (to S.H.); and The Foundation of Finnish Anti-Tuberculosis Association (to S.H.).

**Competing interests:** I have read the journal's policy and the authors of this manuscript have the following competing interests: J.L. is a shareholder of a company (Kubios) that designs ECG and heart rate variability analysis software. Other co-authors declare that they have no conflict of interest. This does not alter our adherence to PLOS ONE policies on sharing data and materials.

## Introduction

Obstructive sleep apnea (OSA) is a prevalent sleep disorder; globally it is estimated that nearly 1 billion adults have OSA and the prevalence can exceed 50% in some countries [1]. Even though OSA is strongly associated with the progression of cardiovascular diseases [2], the electrocardiogram (ECG) signal recorded during polysomnography (PSG) is rarely used. Yet, the nocturnal ECG might provide important information about the risk of cardiac events in OSA patients. In addition to cardiovascular diseases, OSA is considered an independent risk factor for stroke, but it can also be a consequence of a stroke [3]. This bidirectional interaction between OSA and stroke increases the risk of recurrent strokes [4]. This interaction emphasizes the importance of screening stroke patients for OSA to prevent recurrent strokes.

Although the cardiovascular consequences of OSA are not fully understood, certain alterations in cardiovascular function are reported to occur in OSA. For example, OSA causes intermittent hypoxia, arousals, and negative intrathoracic pressure swings [5]. These alterations in cardiovascular function disrupt the sympathetic and parasympathetic balance, causing a shift toward sympathetic predominance [5]. In turn, the alterations in sympathovagal balance may affect ventricular repolarization [6]. Prolonged heart rate corrected QT interval (QTc), a surrogate for ventricular repolarization, is also known to be associated with an increased risk of arrhythmias and sudden cardiac death [7]. Furthermore, QT interval prolongation is common following an acute stroke [8].

Several studies have demonstrated the association between OSA and QTc prolongation [9–12] albeit conflicting results exist [13, 14]. The results regarding the variations of QTc intervals in different sleep stages are also conflicting. While studies show different trends or insignificant changes in QTc interval toward deeper sleep stages in non-rapid eye movement (non-REM) stages, the longest QTc intervals are reported to occur in REM sleep [9, 15, 16]. However, in these studies, variations of QTc intervals were not investigated separately in different sleep stages and during wake [9, 14–16].

Stroke patients are not routinely screened for OSA albeit they have a high risk of having OSA [17]. Currently, if no obvious cause of stroke can be found, stroke patients are screened with 24-h ECG monitoring giving a possibility to detect indirect electrophysiological consequences of OSA in this patient group. Thus, this study aimed to investigate inter-sleep stage variations of ventricular repolarization in OSA patients with previously diagnosed stroke by examining QTc intervals in different sleep stages. We hypothesize that the OSA patients with stroke history have longer QTc intervals compared to the OSA patients without stroke history and that QTc intervals in REM sleep are longer compared to the other sleep stages.

## Methods

### Dataset

The studied dataset is a subpopulation of a large retrospective study comprising over 900 consecutive PSG recordings of suspected OSA subjects. In this study, 18 subjects (13 men) with stroke history fulfilled the inclusion criteria: apnea-hypopnea index (AHI) $\geq$ 15, no ventricular pacing, and total sleep time $\geq$ 4 hours in PSG. The control group comprised 18 (12 men) subjects with similar inclusion criteria but without a previous stroke history. The subjects in the control group were randomly selected using a random number generator ("randi" function in MATLAB R2021b [MathWorks Inc, MA, USA]). The PSG recordings were conducted at the Sleep Disorders Centre, Princess Alexandra Hospital (Brisbane, Australia) during 2015–2017 using the Compumedics Grael acquisition system (Compumedics, Abbotsford, Australia). The

recorded PSGs were manually scored in accordance with the American Academy of Sleep Medicine (AASM) 2012 guidelines [18]. The scoring protocol of PSGs is detailed in our previous studies [19, 20]. The retrospective data collection approval was granted by the Institutional Human Research Ethics Committee of the Princess Alexandra Hospital (HREC/16/QPAH/021 and LNR/2019/QMS/54313). Due to the study's retrospective nature, the need for informed consent was waived by the Metro South Human Research Ethics Committee.

### Analysis of the ECG recordings

The ECG signals were recorded with a sampling frequency of 256 Hz using a modified lead II configuration. Signals were truncated into 30-second segments according to sleep stages. The ECG analysis was performed with Kubios HRV software (Kubios Oy, Kuopio, Finland) [21]. Signals were detrended by the smoothness priors method [22] with a smoothing parameter of 500 and the Pan-Tompkins method [23] was used to detect the R-peaks in the segments. Furthermore, the average beat waveform was computed from each ECG segment after detecting all beats. A QT interval was measured from the beginning of the averaged QRS complex until the end of the T wave. QTc intervals were calculated according to Bazett's formula [24]. ECG segments with $\geq$ 5% beat correction were excluded from the analysis to avoid the inclusion of low-quality segments. A slow heart rate (i.e. around 40–50 beats per minute [BPM]) during sleep is common [25] and can be further lowered due to possible transient decrease during apnea/hypopnea events [26]. However, as the low technical quality of ECG recording can lead to erroneously low HR values in segments due to missed R-peaks detection, we also considered an average HR of 30 BPM as a threshold for the exclusion of low-quality segments.

RR, QT, and QTc intervals were calculated for each ECG segment. Only those wake sections that continued for at least 3 minutes were considered to represent actual wake condition rather than short awakening between sleep stages. To evaluate the changes in the selected ECG parameters between sleep and resting states, the relative changes to the wake stage were also considered. This was done by subtracting the median value of the selected ECG parameter (i.e., RR, QT, or QTc) during the wake from the value calculated during different sleep stages. This procedure was applied for each subject to minimize the individual variation in the baseline measures and compensate for possible systematic errors, thus providing detailed information on the effect of sleep stages on ECG parameters.

The Mann-Whitney U test was used to evaluate the statistical significance of the changes in ECG features between sleep stages. Multiple regression analyses were utilized to analyze the effect of age, sex, body mass index (BMI), AHI, arousal index (AI), stroke history, and sleep stages on QTc intervals to identify possible confounding factors. Four linear regression models were utilized to investigate each sleep stage (N1, N2, N3, REM) independently pairwise with wake. This was done to investigate whether sleep stages have an independent effect on QTc intervals after adjustment for confounding factors. The statistical data analysis was performed with MATLAB R2021b. A $p$-value of $< 0.05$ was considered statistically significant.

### Results

The demographic data of the studied population are presented in Table 1 and the number of analyzed segments in each sleep stage is presented in Table 2.

RR, QT, and QTc intervals showed significant differences across sleep stages in both control and stroke groups (Fig 1), however, absolute values of QT and QTc were significantly ($p<0.05$) higher in the stroke group compared to the control group regardless of sleep stages. The QTc trend showed a steady increase ($p<0.01$) toward deeper sleep after an initial decrease in N1 in both groups (Fig 1).

**Table 1. Demographic data and sleep characteristics of control and stroke groups.** Reported *p*-values are from the Mann-Whitney U test (for continuous variables) or the Chi-squared test (for categorical variables).

| Characteristics | Control (*n* = 18) | Stroke (*n* = 18) | *p*-value |
|---|---|---|---|
| **Age (years)** | 48.3 (46.2, 49.6) | 67.8 (55.6, 73.3) | < 0.001 |
| **Sex, male/female** | 13/5 | 12/6 | 0.76 |
| **BMI (kg/m$^2$)** | 35.1 (31.0, 37.9) | 36.2 (33.8, 43.5) | 0.28 |
| **AHI (events/hour)** | 26.5 (19.6, 35.1) | 30.3 (20.7, 33.6) | 0.60 |
| **AI (events/hour)** | 4.7 (1.7, 8.1) | 8.1 (3.7, 9.8) | 0.35 |
| **TST (minutes)** | 339.7 (321.0, 368.5) | 316.0 (283.0, 341.0) | 0.03 |
| **Sleep efficiency (%)** | 79.6 (74.9, 82.7) | 71.2 (66.7, 77.4) | 0.02 |
| **N1 (% from TST)** | 13.2 (9.3, 17.5) | 14.2 (7.8, 22.4) | 0.85 |
| **N2 (% from TST)** | 49.1 (43.2, 59.0) | 46.5 (42.6, 52.3) | 0.32 |
| **N3 (% from TST)** | 17.5 (10.6, 23.9) | 17.9 (10.7, 26.2) | 0.92 |
| **REM (% from TST)** | 16.9 (13.7, 20.3) | 18.8 (11.9, 24.8) | 0.58 |

All values are presented as median (interquartile range), except sex. Abbreviations: BMI = body mass index, AHI = apnea-hypopnea index, AI = arousal index, TST = total sleep time, REM = rapid eye movement sleep.

Related to RR, QT, and QTc intervals, the magnitude of the changes from the wake stage was significantly higher (*p*<0.001) in the stroke group compared to the control group in all sleep stages (Table 3). The only exceptions were observed in QTc in N1 sleep and REM sleep.

Similar to the absolute changes, RR, QT, and QTc intervals differed significantly across sleep stages while considering the relative changes to the wake stage (Fig 2). The magnitude of QTc changes in relation to the wake stage increased (*p*<0.01) towards deeper sleep and decreased (*p*<0.01) in the REM sleep in both control and stroke groups; the highest (*p*<0.01) changes occurred in the N3 stage. Also, QT increased steadily towards deeper sleep and decreased in REM sleep in both groups, however, these changes were more clearly visible in the stroke group.

Multiple regression analysis indicated that stroke history is significantly associated with QTc interval prolongation in OSA patients after adjustment for age, sex, BMI, AHI, and AI in all sleep stages (Table 4). Furthermore, N3 and REM sleep were associated with QTc interval prolongation compared to the wake stage after adjustment for all confounders (Table 4).

## Discussion

The current study examined changes in RR, QT, and QTc intervals during different sleep stages in OSA patients with and without stroke history. This is, to our knowledge, the first description

**Table 2. The number of analyzed electrocardiography segments in each sleep stage.** The Chi-squared test was applied to test the statistical significance of the segment's distribution between groups.

| Sleep stage | Control (*n* = 18) | Stroke (*n* = 18) |
|---|---|---|
| **Wake (%)** | 1677 (13.0) | 1935 (18.4) |
| **N1 (%)** | 1595 (12.3) | 1428 (13.6) |
| **N2 (%)** | 5655 (43.7) | 3953 (37.6) |
| **N3 (%)** | 2196 (17.0) | 1633 (15.5) |
| **REM (%)** | 1801 (14.0) | 1574 (14.9) |
| **All stages** | 12924 | 10523 |

The number of analyzed segments in each sleep stage was significantly different (*p* <0.05) between the control and stroke groups.

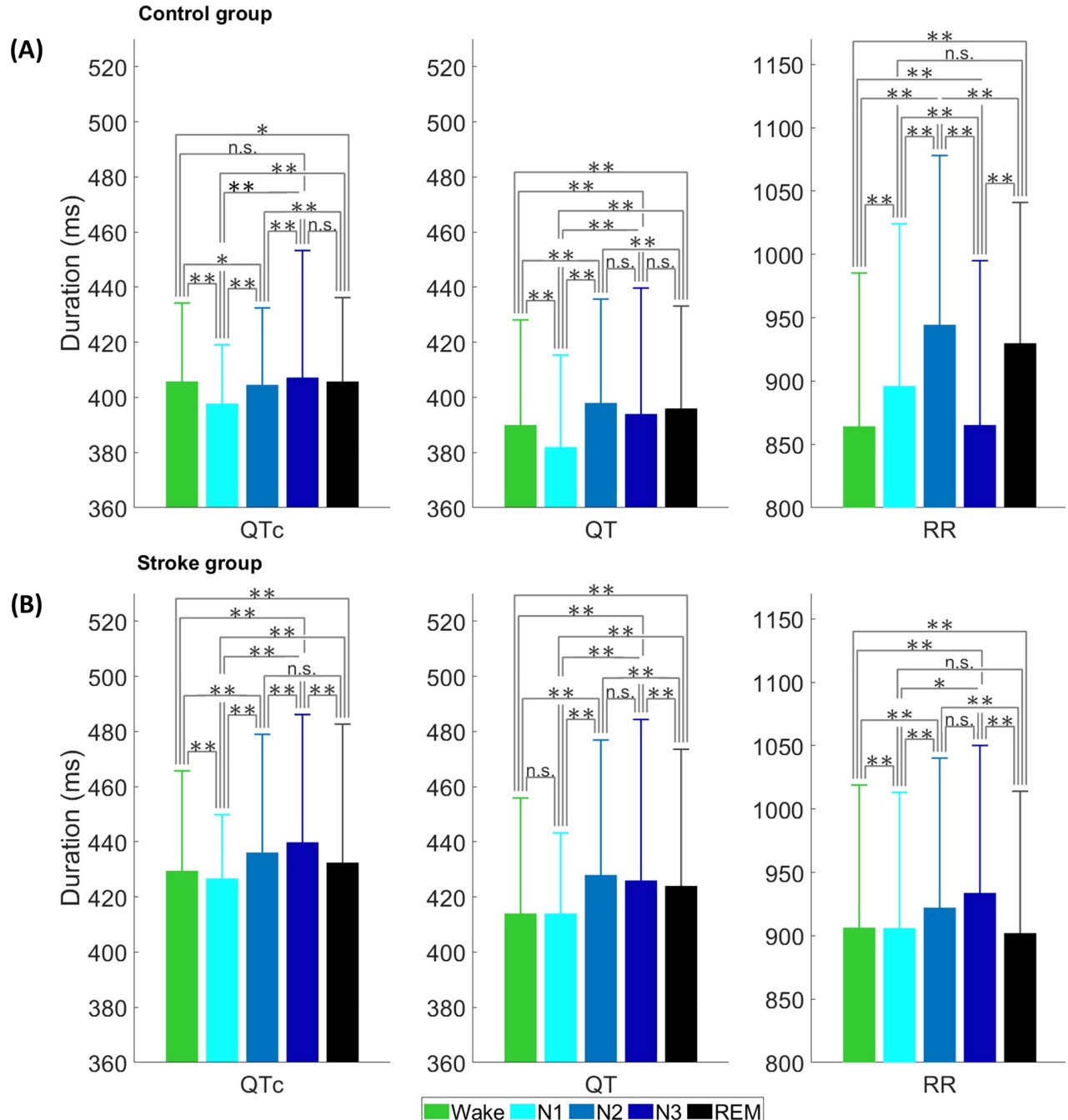

**Fig 1.** Median QTc, QT, and RR intervals across sleep stages: (A) in the control group and (B) in the stroke group. Error bars represent median absolute deviations. $p > 0.05$ = n.s., $p < 0.05 = {}^{*}$, $p < 0.01 = {}^{**}$.

of the inter-sleep stage variations of QTc intervals in patients with OSA and stroke history. Our findings suggest that stroke history is associated with a significant increase in QTc intervals independent of age, sex, BMI, AHI, AI, and sleep stages in OSA patients. As QTc interval prolongation is known to be associated with arrhythmogenicity [7], our results suggest that OSA patients with a stroke history may have an increased risk for ventricular arrhythmias.

**Table 3. Median (interquartile range) of the QTc, QT, and RR interval changes in relation to the wake stage between the stroke and control group.**

|  | ΔQTc (ms) | | | ΔQT (ms) | | | ΔRR (ms) | | |
|---|---|---|---|---|---|---|---|---|---|
|  | **Control** | **Stroke** | **_p_-value** | **Control** | **Stroke** | **_p_-value** | **Control** | **Stroke** | **_p_-value** |
| **N1** | -3.2 (-10.1, 4.1) | 0.2 (-8.0, 9.2) | <0.001 | 7.0 (-2.0, 14.0) | 10.0 (4.0, 22.0) | <0.001 | 39.7 (1.7, 73.5) | 44.3 (10.3, 78.3) | <0.001 |
| **N2** | 1.6 (-7.2, 9.3) | 5.7 (-4.0, 19.0) | <0.001 | 8.0 (-4.0, 18.0) | 12.0 (4.0, 26.0) | <0.001 | 32.5 (-2.5, 77.6) | 41.8 (-1.1, 88.1) | <0.001 |
| **N3** | 6.6 (-7.4, 18.9) | 10.0 (0, 20.9) | <0.001 | 8.0 (-4.0, 20.0) | 14.0 (4.0, 30.0) | <0.001 | 12.6 (-11.6, 44.9) | 42.4 (-10.3, 93.5) | <0.001 |
| **REM** | 3.6 (-6.7, 15.9) | 4.7 (-6.2, 15.4) | 0.150 | 7.0 (-5.0, 18.0) | 10.0 (-2.0, 22.0) | <0.001 | 17.5 (-25.3, 61.2) | 22.7 (-16.1, 81.1) | <0.001 |

In this study, the inter-sleep stage variations of RR, QT, and QTc intervals were studied to gain novel insight of sleep stage-specific cardiovascular risk in OSA patients with and without stroke history. Besides the absolute measures of ECG parameters, the relative change of the parameters from the wake was considered. In both control and stroke groups, QTc intervals increased progressively towards deeper sleep followed by a decrease in REM, the longest QTc intervals being in N3 sleep. In addition, the highest relative changes in ECG parameters were observed in N3 sleep. However, it is noteworthy that the magnitude of changes in QT and QTc intervals were significantly higher in OSA patients with stroke history compared to those without previous stroke history, indicating a higher risk of nocturnal QTc interval prolongation. In addition, regression analysis showed that N3 sleep and REM sleep are independently associated with QTc interval prolongation after adjustment for age, sex, BMI, AHI, AI, and stroke history. The longest QTc intervals were observed in N3 sleep in both groups possibly signaling for elevated risk of arrhythmogenesis during deep sleep.

Several pathophysiological consequences of OSA including hypoxemia, intrathoracic pressure swings, and recurrent arousals lead to autonomic nervous system (ANS) imbalance and may contribute to arrhythmogenesis [27]. Cardiac autonomic nervous activity influences QTc intervals and disruption of autonomic nervous pathways may lead to the prolongation of QTc intervals in healthy subjects [28]. Despite synergies between pathophysiological consequences of OSA, it has been suggested that the sympathovagal imbalance may be the primary cause of cardiac alterations and arrhythmogenesis [27]. Our findings did not support our initial hypothesis that elevated sympathetic activity in REM sleep would lead to longer QTc intervals compared to other stages. Autonomic system alterations in OSA patients consist of parasympathetic activation during respiratory events and sympathetic activation after events, leading to elevated sympathetic activity [27, 29]. Somers et al. showed in contrast to the normal subjects, sympathetic activity reaches high levels during sleep in OSA patients with peak activities during N2 and REM sleep [30]. In addition, Calvo et al. showed that severe OSA may be associated with an increased sympathetic modulation across all sleep stages [31]. Despite the evidence regarding altered autonomic system activity in OSA patients, previous results regarding QTc interval variations in OSA patients are conflicting. Zeng et al. reported a decrease in QTc as sleep gets deeper and an increase in QTc during REM sleep—the highest QTc intervals in REM sleep [9]. Lanfranchi et al. showed insignificant changes in QTc intervals during wake, N2, and N3 sleep and a significant increase in REM sleep [15]. Schmidt et al. reported a slight increase in QTc from the wake stage toward N3 sleep with an insignificant change between N3 and REM [16]. Conversely, our results indicate that QTc intervals increase significantly towards deeper sleep and are the longest in N3 sleep in OSA patients despite stroke history. We assume that by considering the QTc interval changes in relation to the wake stage, patient-specific variations in QTc intervals are compensated and give us new insights to the sleep-specific variations compared to the changes in absolute measures.

Although no studies have focused on altered autonomic system activity in OSA patients with stroke history, evidence of parasympathetic activity and sympathetic decline in post-

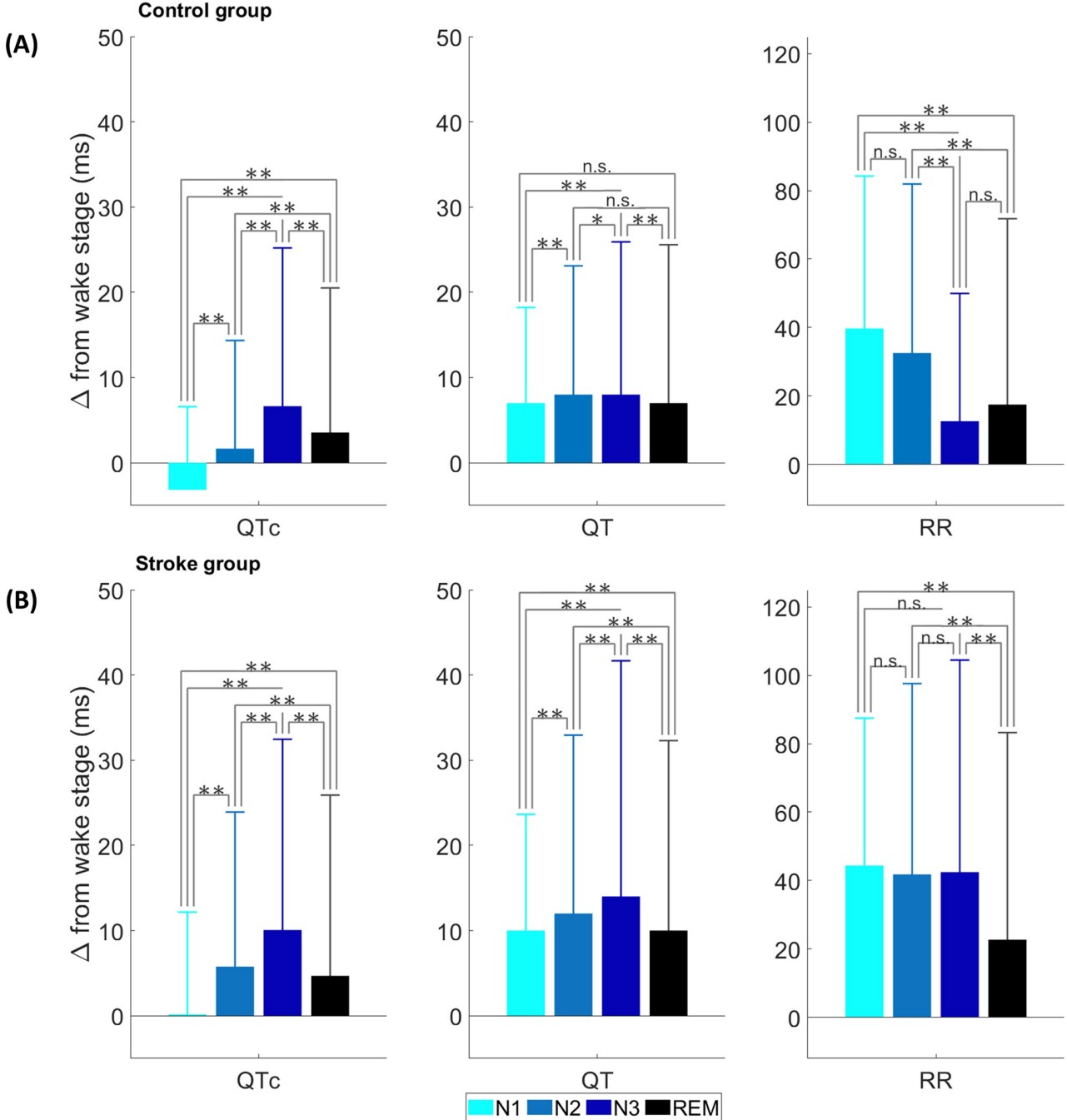

**Fig 2.** Median QTc, QT, and RR interval changes in relation to the wake stage across sleep stages: (A) in the control group and (B) in the stroke group. Error bars represent median absolute deviations. $p > 0.05$ = n.s., $p < 0.05$ = *, $p < 0.0$ 1 = **.

stroke patients has been reported. Brunetti et al. observed that compared to controls, acute stroke patients have a predominant parasympathetic tone during wake and REM sleep accompanied by a reduction of sympathetic tone in REM and parasympathetic tone during N3 [32]. Tobaldini et al. reported predominant vagal modulation and decreased sympathetic modulation across all sleep stages [33]. Based on our observation of RR interval variations as a demonstration of sympathovagal balance, the most notable difference occurs in non-REM sleep

**Table 4. Multiple regression analysis of the effects of confounding factors on QTc intervals in OSA patients.** Except for age and BMI, all parameters are categorical.

| Predictors | N1 Model | | N2 Model | | N3 Model | | REM Model | |
|---|---|---|---|---|---|---|---|---|
| | β-coefficient | *p*-value | β-coefficient | *p*-value | β-coefficient | *p*-value | β-coefficient | *p*-value |
| Age | -0.135 | 0.001 | -0.124 | 0.003 | -0.132 | 0.002 | -0.118 | 0.005 |
| Sex | -13.345 | <0.001 | -13.274 | <0.001 | -13.118 | <0.001 | -13.268 | <0.001 |
| BMI | 0.880 | <0.001 | 0.893 | <0.001 | 0.868 | <0.001 | 0.894 | <0.001 |
| AHI | -0.966 | <0.001 | -0.997 | <0.001 | -0.990 | <0.001 | -0.993 | <0.001 |
| AI | 0.794 | <0.001 | 0.795 | <0.001 | 0.793 | <0.001 | 0.791 | <0.001 |
| Stroke | 34.789 | <0.001 | 34.365 | <0.001 | 34.563 | <0.001 | 34.186 | <0.001 |
| Wake stage | -3.316 | <0.001 | -0.909 | 0.226 | -0.308 | 0.659 | -0.942 | 0.177 |
| Corresponding Sleep stage | -12.143 | <0.001 | 1.035 | 0.109 | 5.842 | <0.001 | 2.749 | 0.001 |

Corresponding sleep stage differs for each model; N1 sleep for the N1 model, N2 sleep for the N2 model, N3 sleep for the N3 model, and REM sleep for the REM model.
Abbreviations: BMI = body mass index. AHI = apnea-hypopnea index, AI = arousal index, REM = rapid eye movement.

specifically in N2 and N3 sleep between OSA patients with and without stroke history. In addition, total sleep time and sleep efficiency were significantly lower in OSA patients with stroke history, indicating the higher amount of wake during the night. These findings are in line with previous results indicating shorter total sleep time and lower sleep efficiency in stroke patients [34] which could be a possible cause of differences in sympathovagal balance between OSA patients with and without stroke history. Despite significant variation in QTc intervals between sleep stages, evidence regarding this distinct dynamicity of QTc intervals in OSA patients is inconclusive. Non-identical characteristics of the ANS in different sleep stages could be one of the causes for non-identical QTc intervals. Furthermore, the dynamicity of QTc intervals could be due to the synergies between the pathophysiological consequences of OSA and their immediate effect on QTc intervals. Recently it was shown that severe desaturation events prolong QTc intervals [35]. Therefore, further research into the effects of type and severity of apneic events on QTc intervals and the effects of pathological consequences of OSA on the ANS could reveal specific characteristics of distinct inter-sleep variation of QTc.

This study was not without limitations. First, the date of stroke occurrence before admitting to the PSGs and the information on the anatomical location of the stroke were not available in our data set. QTc prolongation and cardiac abnormalities have are to be the most prevalent in relation to strokes occurring in the insular region [8], showing the stroke region can affect the duration of QTc intervals. Second, a complete list of patients' medications was not available in our dataset and thus, the possible confounding effect of medications could not be quantified in our work. Third, the OSA patients with stroke history were significantly older compared to patients without stroke history. Being a confounding factor, age had no considerable effect on our results as the estimated effect of age was small in our dataset. Fourth, regression analysis showed both the AHI and AI affect QTc intervals. Therefore, apneic events affect QTc intervals and further studies on the effects of apneic events and their occurrences in different sleep stages could reveal more about their immediate impact on QTc intervals and the risk of arrhythmias. Last, the number of male patients was higher compared to the females in both groups. There is a known association between sex and QTc intervals, females having longer QTc intervals compared to males [36] and this was also seen in the current data. Although sex was found to be a strong confounder in our data, the gender balance in the patient groups involved in this study was not statistically different. As stroke history and sleep stages were demonstrated to be independent factors in QTc interval prolongation, our results indicate that female OSA patients with stroke history are at a higher risk of QTc prolongation in deep sleep compared to males.

## Conclusion

This study shows that the duration of QTc intervals increases progressively towards deeper sleep following by a decrease in the REM sleep in OSA patients with or without stroke history. Furthermore, the stroke history is associated with longer QTc intervals independent of age, sex, BMI, AHI, AI, and sleep stages in OSA patients. These findings indicate that OSA patients with a previous stroke have an elevated risk, especially during deeper sleep, for QTc prolongation—a known risk factor for ventricular arrhythmogenicity and sudden cardiac death.

## Supporting information

**S1 Checklist.**
(PDF)

## Author Contributions

**Conceptualization:** Antti Kulkas, Juha Töyräs, Timo Leppänen, Samu Kainulainen.

**Data curation:** Salla Hietakoste, Samu Kainulainen.

**Formal analysis:** Serajeddin Ebrahimian.

**Funding acquisition:** Timo Leppänen.

**Investigation:** Brett Duce.

**Methodology:** Serajeddin Ebrahimian, Saara Sillanmäki, Samu Kainulainen.

**Project administration:** Samu Kainulainen.

**Resources:** Brett Duce, Jukka A. Lipponen, Samu Kainulainen.

**Software:** Serajeddin Ebrahimian, Jukka A. Lipponen.

**Supervision:** Saara Sillanmäki, Juha Töyräs, Timo Leppänen, Samu Kainulainen.

**Validation:** Saara Sillanmäki, Timo Leppänen, Samu Kainulainen.

**Visualization:** Serajeddin Ebrahimian.

**Writing – original draft:** Serajeddin Ebrahimian.

**Writing – review & editing:** Saara Sillanmäki, Salla Hietakoste, Brett Duce, Antti Kulkas, Juha Töyräs, Timo Leppänen, Jukka A. Lipponen, Samu Kainulainen.

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
