## [Decision Letter · Decision Letter 0]

28 Sep 2022

PONE-D-22-23794Inter-sleep stage variations in corrected QT interval differ between obstructive sleep apnea patients with and without stroke historyPLOS ONE

Dear Dr. Ebrahimian,

Thank you for submitting your manuscript to PLOS ONE. After careful consideration, we feel that it has merit but does not fully meet PLOS ONE’s publication criteria as it currently stands. Therefore, we invite you to submit a revised version of the manuscript that addresses the points raised during the review process.

Your paper was evaluated by an expert in the field and myself. Though the topic is interesting, the reviewer is concerned with some issues regarding analysis. Please read the comments and address the issues accordingly. 

We look forward to receiving your revised manuscript.

Kind regards,

Tomohiko Ai, M.D., Ph.D.

Academic Editor

PLOS ONE

“I have read the journal's policy and the authors of this manuscript have the following competing interests: J.L. is a shareholder of a company (Kubios) that designs ECG and heart rate variability analysis software. Other co-authors declare that they have no conflict of interests.”

Reviewers' comments:

Reviewer's Responses to Questions

**Comments to the Author**

1. Is the manuscript technically sound, and do the data support the conclusions?

Reviewer #1: Yes

2. Has the statistical analysis been performed appropriately and rigorously? 

Reviewer #1: Yes

3. Have the authors made all data underlying the findings in their manuscript fully available?

Reviewer #1: Yes

4. Is the manuscript presented in an intelligible fashion and written in standard English?

Reviewer #1: Yes

5. Review Comments to the Author

Reviewer #1: The present is an observational study

Methods. It should be added how "randomly selected" is defined

Methods. Please report the strobe statement https://www.equator-network.org/reporting-guidelines/strobe/

Methods. It is not cleat how patients were selected. I think that there is a risk of bias toward those patients (probably at high risk) who had all these data available.

Methods. Tables should not be reported in the methods but in the results

Methods.It should be included why HR less than 30 should be defined as low

6. PLOS authors have the option to publish the peer review history of their article (what does this mean?). If published, this will include your full peer review and any attached files.

Reviewer #1: **Yes: **Fabrizio D'Ascenzo

---

## [Author Response · Author response to Decision Letter 0]

14 Nov 2022

Please see the attached Response to Reviewers document.

---

## [Decision Letter · Decision Letter 1]

18 Nov 2022

Inter-sleep stage variations in corrected QT interval differ between obstructive sleep apnea patients with and without stroke history

PONE-D-22-23794R1

Dear Dr. Ebrahimian,

We’re pleased to inform you that your manuscript has been judged scientifically suitable for publication and will be formally accepted for publication once it meets all outstanding technical requirements.

Kind regards,

Tomohiko Ai, M.D., Ph.D.

Academic Editor

PLOS ONE

Additional Editor Comments (optional):

Reviewers' comments:

Reviewer's Responses to Questions

**Comments to the Author**

1. If the authors have adequately addressed your comments raised in a previous round of review and you feel that this manuscript is now acceptable for publication, you may indicate that here to bypass the “Comments to the Author” section, enter your conflict of interest statement in the “Confidential to Editor” section, and submit your "Accept" recommendation.

Reviewer #1: All comments have been addressed

2. Is the manuscript technically sound, and do the data support the conclusions?

Reviewer #1: (No Response)

3. Has the statistical analysis been performed appropriately and rigorously? 

Reviewer #1: (No Response)

4. Have the authors made all data underlying the findings in their manuscript fully available?

Reviewer #1: (No Response)

5. Is the manuscript presented in an intelligible fashion and written in standard English?

Reviewer #1: (No Response)

6. Review Comments to the Author

Reviewer #1: (No Response)

7. PLOS authors have the option to publish the peer review history of their article (what does this mean?). If published, this will include your full peer review and any attached files.

Reviewer #1: **Yes: **Fabrizio D'Ascenzo

---

## [Editor Report · Acceptance letter]

24 Nov 2022

PONE-D-22-23794R1 

Inter-sleep stage variations in corrected QT interval differ between obstructive sleep apnea patients with and without stroke history 

Dear Dr. Ebrahimian:

I'm pleased to inform you that your manuscript has been deemed suitable for publication in PLOS ONE. Congratulations! Your manuscript is now with our production department. 

Kind regards, 

on behalf of

Dr. Tomohiko Ai 

Academic Editor

PLOS ONE